# Longitudinal Profiling of the Macaque Vaginal Microbiome Reveals Similarities to Diverse Human Vaginal Communities

Nicholas S. Rhoades,[a] Sara M. Hendrickson,[b] Danielle R. Gerken,[a] Kassandra Martinez,[a] Ov D. Slayden,[c] Mark K. Slifka,[b] Ilhem Messaoudi[a]

[a]Department of Molecular Biology and Biochemistry, University of California, Irvine, Irvine, California, USA
[b]Division of Neuroscience, Oregon National Primate Research Center, Beaverton, Oregon, USA
[c]Division of Reproductive & Developmental Sciences, Oregon National Primate Research Center, Beaverton, Oregon, USA

Nicholas S. Rhoades and Sara M. Hendrickson contributed equally to this article. Author order was determined by seniority.

**ABSTRACT** The vaginal microbiota plays an important role in women's reproductive and urogenital health. It is now well accepted that a "healthy" vaginal microbiome is dominated by *Lactobacillus* species. Disturbances in this microbial community can lead to several adverse outcomes, including pelvic inflammatory disease and bacterial vaginosis (BV), as well as increased susceptibility to sexually transmitted infections, miscarriage, and preterm births. However, vaginal communities, especially those of women in the developing world, can be comprised of a diverse set of microorganisms in the absence of overt clinical symptoms. The implications of these diverse vaginal microbiomes for women's health remain poorly understood. Rhesus macaques are an excellent translational animal model to address these questions due to significant physiological and genetic homology with humans. In this study, we performed a longitudinal analysis of clinical and microbiome data from 16 reproductive-age female rhesus macaques. At both the taxonomic and functional levels, the rhesus macaque vaginal microbiome was most similar to that of women who harbor a diverse vaginal community associated with asymptomatic/symptomatic bacterial vaginosis. Specifically, rhesus macaque vaginal microbiomes harbored a diverse set of anaerobic Gram-negative bacteria, including *Sneathia*, *Prevotella*, *Porphyromonas*, and *Mobiluncus*. Interestingly, some animals were transiently colonized by *Lactobacillus* and some with *Gardnerella*. Our in-depth and comprehensive analysis highlights the importance of the model to understand the health implications of a diverse vaginal microbiome and test interventions for manipulating this community.

**IMPORTANCE** It is widely accepted that the "healthy" vaginal microbiome of women in the developed world is dominated by *Lactobacillus* species. However, in the developing world, many asymptomatic women harbor diverse vaginal microbial communities that are typically associated with bacterial vaginosis. Many questions remain about the drivers and health implications of a diverse vaginal microbial community. Rhesus macaques provide an excellent translational model to address these questions due to significant physiological and genetic homology with humans. In this study, we performed a longitudinal analysis of clinical and microbiome data from a large cohort of reproductive-age rhesus macaques. At the taxonomic, genomic, and functional levels, the rhesus macaque vaginal microbiome was most similar to that of humans, who harbor a diverse vaginal community associated with asymptomatic/ symptomatic bacterial vaginosis. Our in-depth and comprehensive analysis highlights the utility of macaques as a model to study diverse vaginal community state types and test interventions for manipulating the vaginal microbiome.

**KEYWORDS** microbiome, vagina, bacterial vaginosis, metagenomics, rhesus macaque

Address correspondence to Mark K. Slifka, slifkam@ohsu.edu, or Ilhem Messaoudi, imessaou@uci.edu.

The vaginal microbial community modulates several critical physiological functions and protects the host from infection with pathogenic organisms. The "healthy" vaginal microbiome (VM) in women in the developed world is typically dominated by *Lactobacillus* species, including *L. crispatus*, *L. iners*, *L. johnsonii*, and *L. gasseri*. These microbes are considered keystone microbes that rarely co-occur in individuals (1, 2). Lactobacilli inhibit the growth of other vaginal microbes via lactic acid fermentation, which decreases the pH of the vaginal environment (pH 3.5 to 5.5) and competitively excludes genitourinary pathogens, thereby preventing infection (3). Disruptions of this community have significant implications for women's health (4–6). However, approximately ~20% of women in the developed world and ~50% in the developing world harbor a diverse VM with reduced abundance or absence of lactobacilli and a high abundance of taxa such as *Gardnerella*, *Prevotella*, *Sneathia*, and *Mobiluncus* (1, 4). This diverse community is often associated with a heightened inflammatory state, increased susceptibility to sexually transmitted diseases (notably HIV), and the development of bacterial vaginosis (BV) (7).

BV is the most common vaginal infection in the United States, affecting ~30% of women ages 15 to 44 (8). However, the prevalence of BV can vary widely based on geographic location and racial background (8). BV is caused by bacterial overgrowth, disruption of the commensal microbial community, and a proinflammatory environment (9). While typically asymptomatic, BV can result in discomfort and, more importantly, increased susceptibility to sexually transmitted diseases, preterm birth, and pelvic inflammatory disease (6, 10, 11). The antibiotics metronidazole and clindamycin are the current standard treatment for BV. However, BV recurs in ~50% of women within 12 months of antibiotic treatment (12). Additionally, the production of biofilms and the development of other antimicrobial resistance mechanisms will most likely reduce the effectiveness of antibiotic treatments over time (13, 14). Recently, VM transplants have been shown to be effective in the treatment of intractable BV (15). However, microbiome transfer procedures have recently come under scrutiny due to safety concerns (16, 17).

Another potential intervention is the use of prebiotics as an inexpensive and easily accessible alternative to antibiotic treatment for BV (18, 19) to shift a microbial community to a more "beneficial" state. Prebiotics are molecules and/or nutrients that are metabolically accessible to microbes with the goal of enriching for beneficial microbes and/or depleting undesired microbes. Of interest is the intravaginal application of di- or polysaccharides that are preferentially metabolized and fermented by *Lactobacillus* to produce lactic acid and hydrogen peroxide, thereby generating a low-pH vaginal environment and inhibiting the growth of BV-associated bacteria. Clinical trials are under way to test the use of intravaginal lactose (ClinicalTrials registration no. NTC03878511), glucose (ClinicalTrials registration no. NCT03357666), and glycogen (ClinicalTrials registration no. NCT02042287) to treat BV symptoms. Sucrose in particular has shown potential for improving clinical markers of BV in humans (20) and shifting the VM of rhesus macaques (21). However, the clinical study only examined changes in clinical symptoms and Amsel criteria at one time point after 14 days of treatment and did not interrogate changes in microbial communities (20). The preclinical study with macaques utilized animals with a vaginal microbial community that had >1% *Lactobacillus* and did not identify the *Lactobacillus* species present; nor did it report individual animal data (21). More recently, a study of rhesus macaques reported that maltose can shift diverse vaginal communities into *Lactobacillus*-dominated ones (22). However, this shift was short-lived, and the relative abundance of *Lactobacillus* species was >5% before treatment in the test group (22). Therefore, it is still unclear whether a sucrose intervention could improve clinical outcomes when the frequency of *Lactobacillus* species is extremely low, which *Lactobacillus* strains can respond to sucrose treatment, and how effective this intervention is.

To address these questions, we utilized a combination of clinical tests, 16S rRNA amplicon sequencing, and shotgun metagenomics to characterize the taxonomic and functional landscape of the rhesus macaque VM before and after intravaginal sucrose

**TABLE 1** Clinical and microbial characteristics of screening animals

| Clinical parameter or microbial characteristic | Result for macaque: | | | | | | | | | |
|---|---|---|---|---|---|---|---|---|---|---|
| | RM1 | RM2 | RM3 | RM4 | RM5 | RM6 | RM7 | RM8 | RM9 | RM10 |
| Clinical parameters | | | | | | | | | | |
| Nugent score | 9 | 9 | 8 | 5 | 10 | 8 | 7 | 2 | 5 | NC[a] |
| >20% clue cells | Yes | Yes | Yes | No | Yes | Yes | Yes | No | No | NC |
| Vaginal pH | 7 | 7.5 | 5.5 | 7.5 | 7 | 7.5 | 5.5 | 6 | 4.5 | 8 |
| | | | | | | | | | | |
| 16S amplicon relative abundance | | | | | | | | | | |
| *Leptotrichiaceae* (*Sneathia*) | 27.26 | 21.26 | 30.52 | 89.78 | 9.7 | 9.37 | 0.36 | 2.78 | 2.13 | 0.05 |
| *Lactobacillus* | 0.02 | 0 | 0 | 0.13 | 0.03 | 0.03 | 42.98 | 55.75 | 84.44 | 0.1 |
| *Prevotella* | 14.84 | 5.02 | 13.58 | 0 | 7.46 | 8.41 | 4.97 | 0.25 | 0.02 | 0.02 |
| *Mobiluncus* | 10.53 | 14.97 | 3.07 | 0 | 7.01 | 8.16 | 0.46 | 0.62 | 0.01 | 0 |
| *Gardnerella* | 0 | 0 | 0 | 0 | 1.17 | 0 | 0 | 6.01 | 8.29 | 0 |
| *Porphyromonas* | 6.66 | 10.78 | 11.08 | 0.29 | 2.95 | 5.18 | 0.32 | 0.36 | 0 | 8.7 |
| *Trichococcus* | 0.31 | 2.18 | 1.1 | 0.18 | 1.18 | 0.62 | 10.76 | 0.67 | 0.01 | 61.01 |
| *Campylobacter* | 3.74 | 3.79 | 7.1 | 0 | 5.74 | 15.94 | 0.07 | 0.27 | 0 | 0.02 |
| *Anaerococcus* | 0.43 | 0.07 | 1.75 | 0.35 | 0.27 | 0.37 | 15.77 | 0.75 | 0.11 | 15.1 |
| *Catonella* | 0 | 0 | 0 | 0 | 8.91 | 9.87 | 0 | 0.14 | 0 | 0 |
| *Fusobacterium* | 1.71 | 0.3 | 16.22 | 0.02 | 0 | 0 | 0.01 | 0.11 | 0.01 | 0 |
| *Peptoniphilus* | 4.14 | 3.81 | 1.62 | 0.24 | 1.56 | 3.91 | 0.27 | 0.58 | 0.01 | 1.7 |

[a]NC, not collected.

treatment. Additionally, we determined the relatedness of rhesus macaque and human vaginal microbes by whole-genome resolution. Previous studies have defined the taxonomic landscape of pigtail (23), cynomolgus (24), and rhesus (25) macaque VM. These earlier studies have shown that macaques harbor a diverse vaginal community (23–25) that shares taxa with the diverse community state type associated with BV in humans (24). These patterns include a low abundance of *Lactobacillus* spp. and a high abundance of *Sneathia*, *Prevotella*, and *Mobiluncus*, among others. However, these previous studies have relied on amplicon sequencing techniques that are limited in resolution and did not address the functional potential of the macaque vaginal community or examine this similarity at a whole-genome level. The data presented herein further strengthen the case for using macaques to understand the drivers and health implications of a diverse vaginal microbiome and to test interventions for manipulating community state types.

## RESULTS

**Rhesus macaques display the clinical hallmarks of BV and are colonized by a diverse vaginal microbiome.** To determine if rhesus macaques can serve as a model of BV, we screened 9 reproductive-age female rhesus macaques. Of these animals, six (66%) had a Nugent score of >7 and the presence of clue cells (Table 1). Additionally, 5 of these 6 animals had a vaginal pH of ≥6.0 (Table 1). These data suggest that the majority of rhesus macaque females display the clinical hallmarks of BV. Profiling of the vaginal microbial (VM) communities using 16S rRNA gene amplicon sequencing showed a high relative abundance of bacterial taxa associated with BV, including *Sneathia*, *Mobiluncus*, *Prevotella*, and *Gardnerella* (Table 1). Interestingly, 3 animals (33%) included in this screen had a high relative abundance of *Lactobacillus* (43 to 84%). These data suggest that a subset of rhesus macaques can naturally harbor a *Lactobacillus*-dominated vaginal community.

**Intravaginal sucrose treatment does not alter the vaginal microbiome of rhesus macaques.** We screened an additional 20 animals for inclusion in a longitudinal study to determine the efficacy of vaginal sucrose gel to improve BV clinical indicators and increase the abundance of *Lactobacillus* in the VM. Initial screening of these 20 animals indicated that 4 animals did not display the clinical indicators of BV (pH <5 and/or Nugent score of <4) and were excluded from the longitudinal study. The remaining 16 animals were split evenly into two groups. Animals in group 1 received seven daily applications of a sucrose gel vaginally, while animals in group 2 received seven daily

applications of the gel alone. Since hormonal levels can influence VM composition (24, 26, 27), animals were administered an oral contraceptive for 21 days (1 month prior to sucrose treatment) to synchronize the menstrual cycles of the animals (Fig. 1A). Contraceptive treatment resulted in a significant decrease in the levels of progesterone ($P_4$) and estradiol ($E_2$) (Fig. 1B and C). Seven days after cessation of the contraceptive treatment, the animals were then treated with either sucrose or placebo gel intravaginally for 7 days. These animals were sampled at 5 additional time points post-sucrose/placebo treatment across 35 days, to collect clinical (vaginal pH, Nugent scores, whiff test, and clue cells), circulating hormone levels (progesterone and estradiol), and microbiome (16S amplicon sequencing) data (Fig. 1A).

Sucrose treatment did not alter clinical measurements (pH, Nugent score, whiff test, or presence of clue cells) or levels of progesterone or estradiol up to 28 days after treatment (see Fig. S2A to F in the supplemental material). Additionally, we observed no differences in VM diversity or overall community composition (see Fig. S3A to C in the supplemental material). Sucrose treatment also did not increase the relative abundance of the genus *Lactobacillus* (Fig. S3D). Since sucrose treatment did not lead to measurable changes in clinical or microbiome measurements, we combined the two groups to generate a longitudinal data set to further our understanding of rhesus macaque VM community dynamics.

Analysis of the clinical markers of BV in all 16 animals showed an association with menstrual cycle. Specifically, menses were associated with reduced Nugent scores and a smaller proportion of animals that met all three measured Amsel criteria while pH levels increased (Fig. 1D to F). The association between menstrual cycle and these clinical markers is more evident when comparing Nugent scores and vaginal pH in animals with or without menses (Fig. 1G and H).

**The taxonomic landscape of the rhesus macaque vaginal microbiome.** Using 16S rRNA gene amplicon sequencing, we characterized the taxonomic landscape of the rhesus VM across all eight time points (Fig. 2A). The rhesus VM is a low-diversity community with an average of ~50 amplicon sequence variants (ASVs) per sample across all time points (Fig. 2B). Interestingly, the overall community composition remained stable across time points in 7 animals (Fig. 2C). These "stable" communities were dominated by either *Prevotella*, *Porphyromonas*, or *Sneathia* and were associated with a higher vaginal pH (Fig. 2A and C). The VM of the remaining 9 animals was more variable, transitioning between states and communities dominated by microbes that were less commonly observed within our study population, such as *Gardnerella* and *Lactobacillus* (Fig. 2D). Finally, we found that individual (permutational multivariate analysis of variance [PERMANOVA], $P = 0.009$) rather than time point (PERMANOVA, $P = 0.42$) was the best predictor of community composition (see Fig. S4A and B in the supplemental material).

We next explored the most abundant taxa within the rhesus macaque VM. A small number of samples (15/112) were dominated (>50% relative abundance) by a single microbe (Fig. 2A), including 9 samples dominated by *Sneathia*, 2 by *Gardnerella*, and 4 by *Lactobacillus*. The remaining samples contained communities composed of multiple anaerobic bacteria, including *Sneathia*, *Porphyromonas*, *Prevotella*, *Fastidiosipila*, *Catonella*, *Mobiluncus*, and *Atopobium* (Fig. 2A). Additional taxa found in lower abundance across all samples include *Parvimonas*, *Dialister*, *Fusobacterium*, *Treponema*, *Peptoniphilus*, and *Campylobacter* (Fig. 2A). The relative abundance of the most abundant 25 taxa did not cluster by time of sample collection or menstruation status; however, samples from some animals with a "stable" VM did cluster (Fig. 2A). Interestingly, *Lactobacillus* was detectable in 51 samples and found in a relative abundance above 30% in 5 samples. These five samples came from two monkeys across 4 time points (Fig. 2A).

We next explored the relationship between clinical measures, hormone levels, and the relative abundance of vaginal microbes using repeated-measure correlations (Fig. 2E). As expected, the relative abundance of *Lactobacillus* was negatively correlated with both vaginal pH and Nugent score (Fig. 2E). In contrast, the relative abundance of

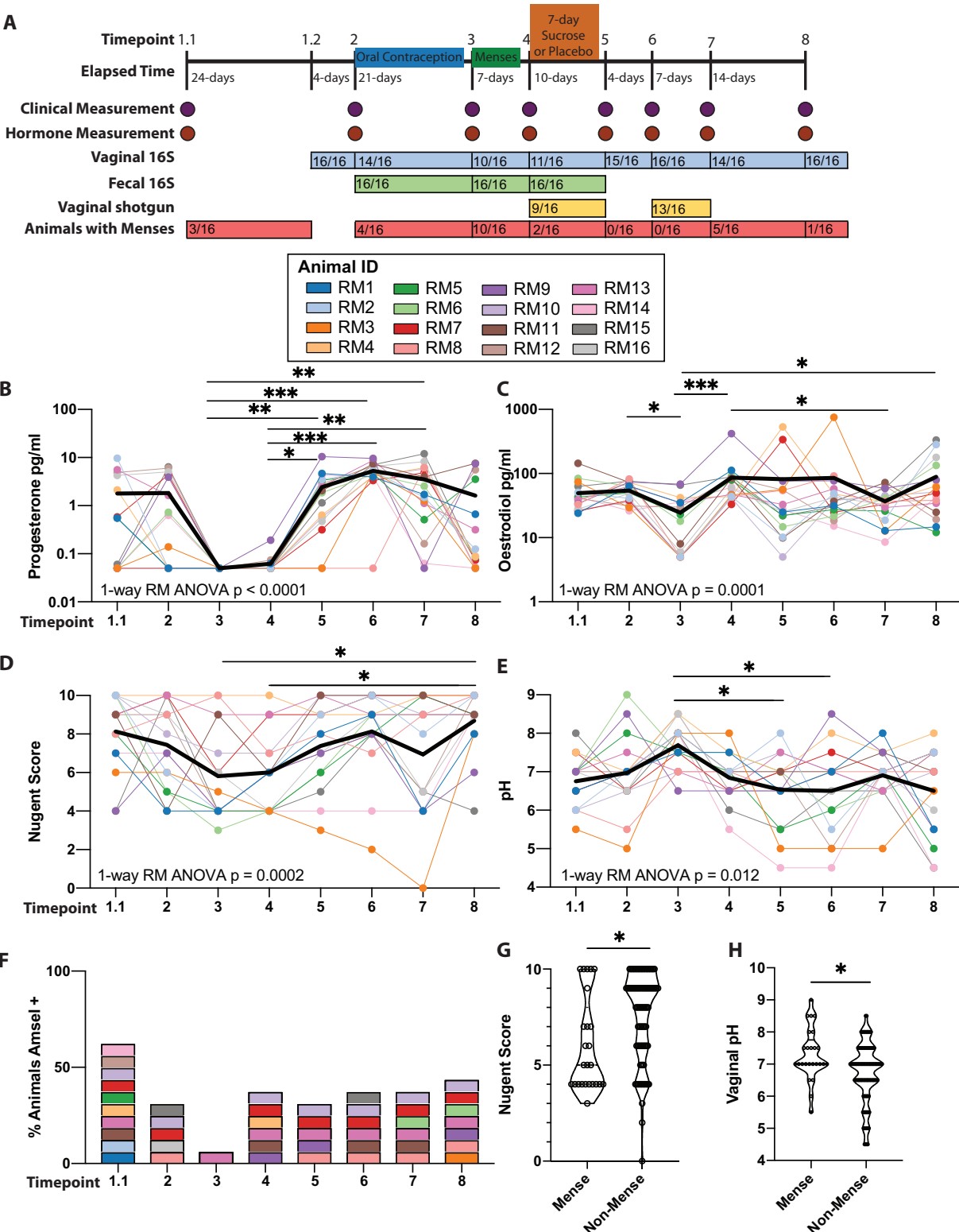

**FIG 1** Longitudinal assessment of clinical markers associated with BV. (A) Study design and timeline. (B to E) Scatterplot of measured systemic (B) progesterone, (C) estradiol, (D) Nugent scores, and (E) pH. Each dot represents an individual sample, with solid lines connecting samples from the same individual across time. The bold black line represents the mean value across time. Significance was measured by nonparametric one-way repeated-measure ANOVA (Friedman test) with Dunn's *post hoc* comparisons between time points. The overall *P* value is denoted on each graph, and bars denote significance of *post hoc* tests: *, $P < 0.05$; **, $P < 0.01$; and ***, $P < 0.001$. (F) Stacked bar graph denoting the percentage of animals that meet Amsel criteria. Colors correspond to those used in panels B to E. (G and H) Scatterplot of (G) Nugent scores and (H) vaginal pH comparing samples collected in animals with and without menses. Significance was determined by nested *t* test within an individual: *, $P < 0.05$.

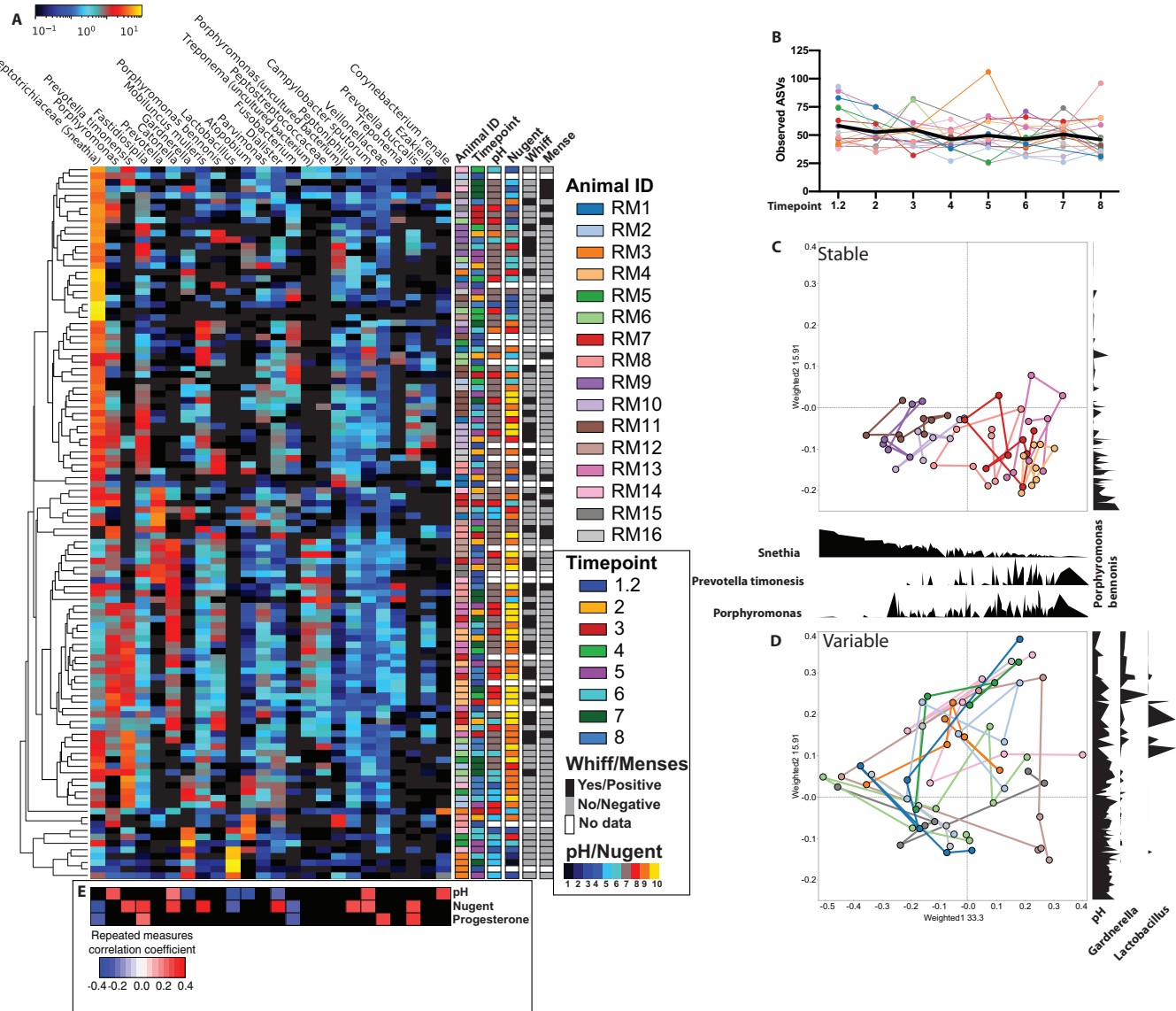

**FIG 2** Longitudinal changes in the rhesus macaque vaginal microbiome. (A) Heat map of the 25 most abundant taxa across all samples ordered from left to right by average abundance. Samples clustered by average linkage of Bray-Curtis distance between samples, as illustrated by the vertical tree. Shown are metadata associated with each sample, including animal ID, time point, Nugent score, vaginal pH, whiff test positivity, and menstruation status. (B) Scatterplot of absolute sequence variants (ASVs) at each time point. Each dot represents an individual sample, with solid lines connecting samples from the same individual across time. The bold black line represents the mean value across time. (C and D) Principal-coordinate analysis of weighted UniFrac distance colored by individuals with lines connecting samples collected from the same individual over time, with density plots of key microbial taxa along the PCoA1 and PCoA2 axis. (E) Heat map of repeated-measure correlation values between vaginal pH, Nugent scores, and systemic progesterone levels against the relative abundance of the top 25 microbes. Significant ($P < 0.05$) positive correlations are shaded in red, negative in blue, and nonsignificant ($P > 0.05$) in black.

*Sneathia*, the most abundant microbe in our data set, was negatively correlated with Nugent score and progesterone levels but not vaginal pH (Fig. 2E; Fig. S4C and D). Additionally, the abundances of *Prevotella timonensis*, *Mobiluncus mulieris*, and *Campylobacter sputorum* were also positively correlated with Nugent scores (Fig. 2E). Levels of estradiol did not correlate with clinical markers or relative abundance of specific microbes. Surprisingly *Gardnerella* was also negatively correlated with vaginal pH (Fig. 2E). However, the vaginal pH of those animals was in the 5 to 7 range, and those animals also harbored a low abundance of *Lactobacillus*.

**The rhesus macaque vaginal microbiome is comparable to human non-*Lactobacillus*-dominated vaginal communities.** Next, we compared our longitudinal 16S rRNA gene amplicon data to those reported in humans by Gosmann et al. (4). This

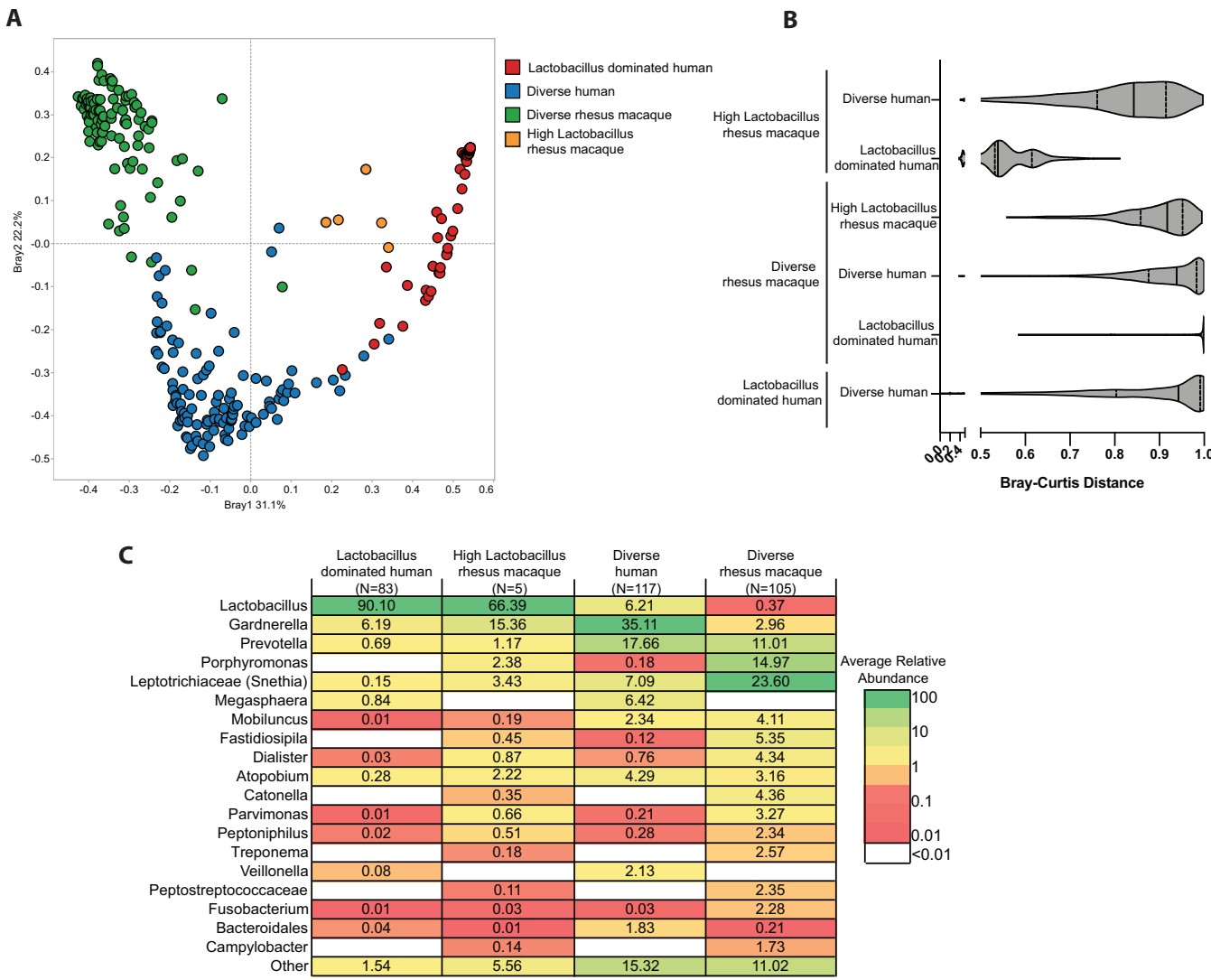

**FIG 3** Comparison of human and rhesus macaque vaginal microbiomes. (A) Principal-coordinate analysis of Bray-Curtis distance between rhesus macaque vaginal microbiome, *Lactobacillus*-dominated human vaginal microbiome, and diverse non-*Lactobacillus*-dominated human vaginal microbiome samples. (B) Violin plot of the average Bray-Curtis distance between the rhesus macaque vaginal microbiome and the two representative human communities. (C) Table of shared and exclusive bacterial genera between the three microbial community types. To be included in this analysis, genera had to be present in 10% of samples within a group at <0.1% relative abundance.

study was selected for the large number of vaginal samples examined and the presence of both *Lactobacillus*-dominated (n = 83) and diverse (n = 117) communities. Rhesus macaque samples were divided into "diverse" (n = 107) and "high *Lactobacillus*" (n = 5). Due to differences in the methodology, a comparison could only be made qualitatively at the genus level. A principal-coordinate analysis (PCoA) shows that the "diverse" rhesus VM was more similar to "diverse" than "*Lactobacillus*-dominated" human vaginal communities (Fig. 3A and B). Six bacterial genera were shared between at least one human and one macaque group with ≥1% relative abundance: *Lactobacillus*, *Gardnerella*, *Prevotella*, *Sneathia*, *Mobiluncus*, and *Atopobium* (Fig. 3C). *Megasphaera*, *Veillonella*, and *Bacteroidales* were only found in the diverse human vaginal microbiome (Fig. 3B). On the other hand, *Porphyromonas*, *Fastidiosipila*, *Dialister*, *Parvimonas*, *Catonella*, and *Campylobacter* were only found in the "diverse" macaque VM (Fig. 2B).

**Metagenomic genome assembly reveals taxa in the rhesus macaque vaginal microbiome similar to human urogenital bacteria.** Shotgun metagenomics provides higher-resolution taxonomic information and functional potential of microbial

communities not attainable using 16S rRNA amplicon sequencing. Shotgun metagenomic libraries were prepared from vaginal samples collected before and 7 days after sucrose/placebo treatment (Fig. 1A). We eliminated any samples with less than 1 million reads after host decontamination, resulting in the loss of 8 samples from the pretreatment and 3 samples from the posttreatment time points (see Fig. S5A in the supplemental material). As noted for 16S rRNA amplicon sequencing, sucrose treatment did not exert a significant impact on the functional potential of the VM (Fig. S5C). Although we were unable to identify the lactobacilli colonizing the rhesus VM at the species level using 16S amplicon sequencing or metagenomic genome assembly, annotation of shotgun metagenomic reads using MetaPhlan2 revealed the presence of the lactobacilli *Lactobacillus johnsonii*, *L. amylovorus*, and *L. acidophilus* (Fig. S5B).

We employed metagenomic genome assembly to generate a higher-resolution picture of which bacterial taxa were present in the rhesus macaque VM and how they relate to the genomes of bacterial strains isolated from the human VM. We constructed a total of 78 metagenomically assembled genomes (MAGs) with >80% genome completeness and <2% contamination, as measured by CheckM (see Table S2 and Fig. S5D in the supplemental material). The MAGs were largely representative of dominant taxa identified in our 16S rRNA amplicon sequencing data, including 9 *Gardnerella*, 6 *Mobiluncus*, 8 *Prevotella*, 2 *Campylobacter*, and 3 *Sneathia* genomes (Table S2 and Fig. S5D). We next determined the relationship between our MAGs and human isolates. The macaque *Mobiluncus* and *Sneathia* genomes were most closely related to the common human bacteria *Mobiluncus mulieris* and *Sneathia sanguinegens*, respectively (Fig. 4A and B). The macaque *Gardnerella* genomes were most closely related to *Gardnerella vaginalis*, commonly detected in non-*Lactobacillus*-dominated communities and often associated with BV in humans (Fig. 4C). Additionally, assembled vaginal *Prevotella* and *Campylobacter* genomes were distinct from those we previously assembled from rhesus macaque fecal samples (Fig. 4D and E). Our assembled *Prevotella* genomes were most closely related to *Prevotella timonensis* and *Prevotella buccalis* previously isolated from human vaginal samples (Fig. 4D). The assembled *Campylobacter* genomes were most similar to *Campylobacter sputorum*, which is most commonly associated with livestock urogenital samples (Fig. 4E).

**The rhesus macaque vaginal microbiome is functionally more similar to women with a diverse vaginal microbiome.** To determine the functional potential of the rhesus VM compared to that of the human VM, we compared our shotgun metagenomic data to those obtained from clinical studies by Oliver et al. (28) and Lev-Sagie et al. (15). These studies analyzed samples from human vaginal communities that were classified as "*Lactobacillus* dominated" (>90% *Lactobacillus*), "diverse asymptomatic" (<90% *Lactobacillus*), and "recurrent BV." We used supervised random forest modeling to determine if the overall functional capacity of the VM could be used to distinguish between samples collected from asymptomatic women (either "*Lactobacillus* dominated" or "diverse asymptomatic"), women with BV, and rhesus macaques. The overall random forest model was 83% accurate at classifying samples into the four groups, with overlap between the VMs from women with BV and asymptomatic women with a diverse VM (Fig. 5A). The VM of rhesus macaques was more closely related to that of asymptomatic women with diverse communities than to the VM of women with BV (Fig. 5B).

We then extracted the functional Gene Ontology (GO) terms that distinguished the four groups (Fig. S5E). Several pathways were unique to each group. For example, pathways associated with palmitate biosynthesis ("palmitoyl-[acyl-carrier-protein] hydrolase activity") and iron acquisition ("heme transport") were common in "*Lactobacillus*-dominated" human VM communities but absent in vaginal communities from rhesus macaques and from asymptomatic women with diverse communities or with recurrent BV (Fig. 5C). Other GO terms, such as "D-gluconate metabolic process" and "UDP-glucose metabolic process," which are associated with carbohydrate and central metabolism, were found in human vaginal communities regardless of health status but absent in the rhesus VM (Fig. 5C). Some GO terms were shared between *Lactobacillus*-

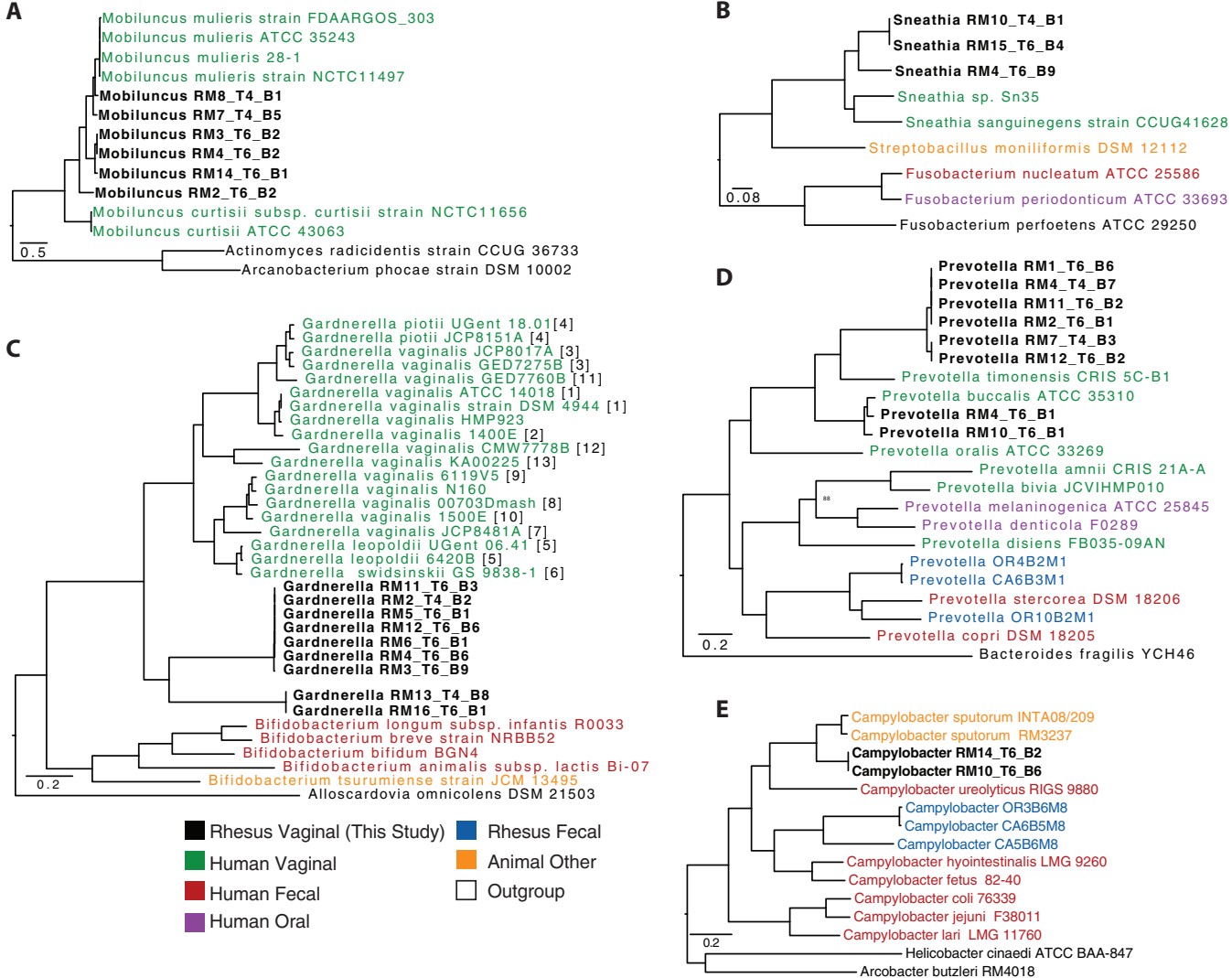

**FIG 4** Genomes assembled from the rhesus macaque vaginal microbiome. Phylogenetic tree built using 500 randomly selected conserved cross-genus gene families (PGfams) for assembled genomes from (A) *Mobiluncus*, (B) *Sneathia*, (C) *Gardnerella*, (D) *Prevotella*, and (E) *Campylobacter*. Genomes in boldface were assembled in this study. Genome IDs are colored by host source. (C) For *Gardnerella*, numbers in brackets indicate the genomic species of each isolate as defined by Vaneechoutte et al. (38), when available.

dominated and diverse communities from asymptomatic women, but absent in women with BV and rhesus macaques, such as "protein-N(PI)-phosphohistidine-sugar phosphotransferase activity" and "phosphoenolpyruvate-dependent sugar phosphotransferase system." Both of these pathways are key for the uptake of sugars by bacteria in the vaginal microbiome.

We also found that the "oxidation-reduction process" pathway, a potential indicator of redox imbalance, was abundant in VM from rhesus and women with diverse vaginal communities but not in *Lactobacillus*-dominated human vaginal communities. Finally, we also identified GO terms that were unique to the rhesus macaque vaginal microbiome, including "antioxidant activity," "arginine catabolic process," and "regulation of pentose phosphate shunt" (Fig. 5C). These pathways are associated with oxidative stress, amino acid metabolism, and central metabolism, respectively.

## DISCUSSION

Several studies conducted in Africa, Asia, and South America have reported that a large fraction of women living in the developing world have a diverse vaginal microbiome defined by the absence or low abundance of Gram-positive *Lactobacillus*

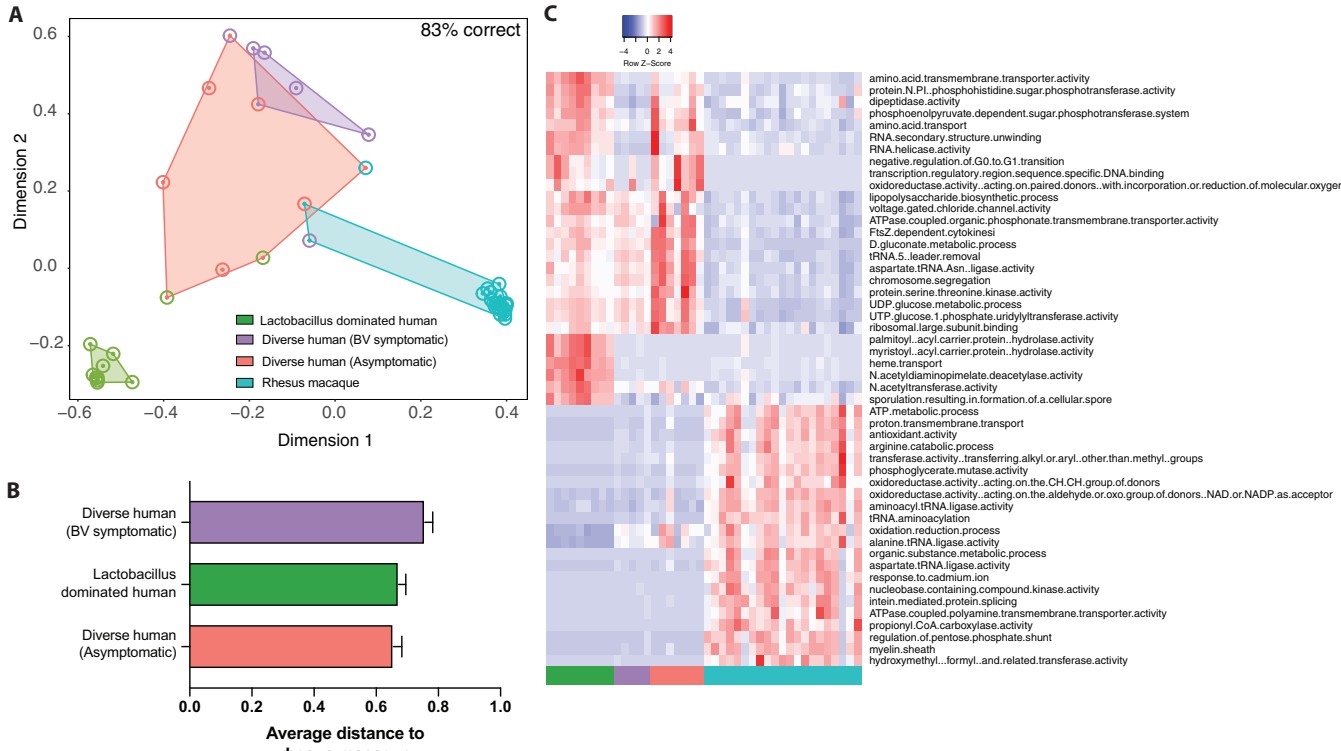

**FIG 5** Comparison of the functional potentials of the rhesus and human vaginal microbiomes. (A) NMDS plot colored by sample source. Outer circles denote if a sample was misclassified at any point during the random forest model generation. The color of the outer circle represents which group that sample was misclassified as. (B) Bar graph of the average random forest proximity between the rhesus macaque vaginal microbiome and the three human communities. (C) Heat map of the 50 most important GO terms as predicted by random forest modeling.

species that typically dominate the human vaginal microbiome in developed countries (1, 2, 4, 11, 28–31). Instead, this diverse community is composed of Gram-negative anaerobic bacteria such as *Gardnerella*, *Prevotella*, *Atopobium*, *Sneathia*, and *Mobiluncus* (1, 29–31). The prevalence of this diverse community state has been connected to multiple host factors, including ethnicity (32), genetics (33), and sexual habits (34). These diverse vaginal communities are largely considered to be a dysbiotic state of the vaginal microbiome, which can result in increased incidence of BV and enhanced susceptibility to sexually transmitted diseases (4, 35). However, important questions about the implications of this community remain unanswered since many women are asymptomatically colonized by these diverse communities. These questions are challenging to ask in the clinical setting and require the availability of a translational animal model that adequately recapitulates the complexity of the human vaginal microbiome.

In this study, we carried out a comprehensive analysis of the rhesus macaque VM using a combination of 16S rRNA amplicon as well as shotgun metagenomic sequencing. We also measured key clinical parameters associated with BV. Our analysis showed that most animals displayed clinical and microbial markers of BV—notably high vaginal pH and high Nugent scores. At the 16S rRNA gene amplicon level, the rhesus VM harbors a diverse set of anaerobic microbes, including *Sneathia*, *Prevotella*, *Mobiluncus*, and *Fusobacterium*, that are often associated with BV. Although the rhesus macaque VM shares a core set of highly abundant genera with humans, it also harbors a diverse set of anaerobic genera less commonly found in human vaginal samples, including; *Porphyromonas*, *Fastidiosipila*, *Catonella*, *Peptostreptococcaceae*, *Spirochaetaceae*, and *Campylobacter* (36, 37).

Our observations are in line with previous studies that have reported a high prevalence and abundance of these genera in other macaque species (23–25). However, our study is the first to report the presence of *Gardnerella*, a key pathogen in human BV and an

indicator of a dysbiotic community (23–25). Specifically, *Gardnerella* was present in 62 of 112 samples, with a relative abundance ranging from 10% to 66% in 15 of those samples. Additionally, using shotgun metagenomics, we assembled 9 *Gardnerella* genomes from 9 different animals that despite considerable similarity were distinct from any previously identified human vaginal *Gardnerella* species, including all 13 recently identified genomic species (38). We observed a similar pattern with other MAGs from *Prevotella*, *Sneathia*, and *Mobiluncus*, in which our assembled genomes were most closely related to human vaginal microbes but formed their own distinct clade. This may indicate host-type-specific adaptions by vaginal microbes. Further comparative genomic and culture studies for each of these genera are warranted to better understand this evolutionary pattern.

As previously reported (25), we found that *Lactobacillus* spp. were largely absent in the rhesus macaque vaginal microbiome. However, a caveat of this study is that we excluded 4 of 20 animals screened for the longitudinal study due to a lack of clinical BV symptoms. These excluded animals were more likely to have a *Lactobacillus*-dominated microbiome. Additionally, in our initial screen, a *Lactobacillus*-dominant microbiome was found in 3 of the 10 animals. In future studies, these animals should be targeted to better understand the stability and drivers of *Lactobacillus* colonization in rhesus macaques. Despite our initial exclusion of animals that lacked clinical hallmarks of BV, a *Lactobacillus*-dominant vaginal community (33 to 87% relative abundance) was transiently observed in animal RM14 at one time point and animal RM3 at four time points. *Lactobacillus* was also detected, with a relative abundance ranging from 1% to 15% in 13 samples across 7 animals. In humans, *Lactobacillus* spp. are the major producers of lactic acids, which results in low vaginal pH (~3.5). In accordance with those observations, the relative abundance of *Lactobacillus* in these animals was negatively correlated with vaginal pH and Nugent scores at these time points. Our shotgun metagenomic analysis revealed the presence of *L. johnsonii*, *L. amylovorus*, and *L. acidophilus* from short-read annotation. While these lactobacilli are closely related to *L. crispatus* and *L. iners* (39), we were unable to assemble a quality *Lactobacillus* MAG and therefore unable to determine the true relationship between rhesus and human vaginal *Lactobacillus* strains. Future studies should analyze vaginal communities from female macaques with a low vaginal pH to isolate and/or assemble *Lactobacillus* genomes for comparison to human strains at the whole-genome level, as we have done for other vaginal taxa.

Previous studies reported that the macaque VM is highly variable over time and that some of this variability was correlated with hormonal cycling (24). We observed a similar trend with multiple taxa being correlated with progesterone levels, including *Mobiluncus mulieris* and *Peptoniphilus*. In contrast to previous longitudinal studies, we found that the VM of some animals remained relatively stable, while that of others was variable over time. While further studies are needed to determine the causes and consequences of these two unique communities' states, one potential explanation for this pattern is fecal contamination. Indeed, some samples obtained from 6 of the 9 animals with "variable" communities had fecal contamination, while samples from only 1 of 7 animals with "stable" communities exhibited fecal contamination. This may suggest that "variable" VMs are responding to community perturbation caused by infiltration of fecal material and microbes. It is also possible that we did not sample often enough to capture the true variability of the rhesus VM. While the human vaginal microbiome is considered to be stable in both pregnant and nonpregnant women (40, 41), others have found that a subset of women have a more temporally variable community (2) and that events such as childbirth can lead to a shift in the vaginal microbiome (42). More recent studies have also observed daily variability has been observed in human samples and was associated with diet, exercise, and hormonal cycling (43). Together, these findings suggest that the perceived stability of the microbiome is affected by both time scale and study population. Rhesus macaques may provide a well-controlled model to further study this pattern.

We also compared the functional capacities of the rhesus VM as well as three distinct human VM communities: "*Lactobacillus* dominated," "asymptomatic diverse," and "recurrent BV." This analysis revealed that the rhesus macaque VM and human VM are functionally distinct, with all three human VM communities functionally enriched in a variety of carbohydrate metabolism GO terms that were not found in the rhesus macaque vaginal microbiome. Host-derived glycogens are the major energy source in the vaginal microbiome and the driver of *Lactobacillus* dominance (44). Differences in host glycogen production between humans and other mammals are hypothesized to contribute to the unique *Lactobacillus*-dominated communities found in humans (7, 45). Although a small percentage of animals can transiently harbor a *Lactobacillus*-dominated VM, microbial communities in rhesus macaques have significantly less vaginal glycogen than those in humans, which may contribute to their inability to sustain a *Lactobacillus*-dominated vaginal community (46). Nevertheless, the rhesus VM is functionally most similar to that of asymptomatic women with a diverse vaginal community.

We tested an intravaginal prebiotic sucrose intervention to drive the vaginal microbiome to a *Lactobacillus*-dominated state and improve clinical markers of BV. Despite previously reported positive results for this intervention (20, 21), we found no clinical or microbial indications of improvement. In our hands, intravaginal sucrose gel did not lower vaginal pH, Nugent scores, or Amsel criterion positivity. Additionally, we did not observe an increase in the relative abundance of *Lactobacillus* or a shift in overall community composition. This intervention could have failed because the relative abundance of *Lactobacillus* in these particular animals was below detection at the start of the intervention. Moreover, shotgun metagenomics indicated that the "protein-N(PI)-phosphohistidine-sugar phosphotransferase activity" pathway, a key step in the bacterial import of sucrose, was largely absent in VM of rhesus macaques as well as women with BV. A microbial community lacking genes within this pathway is less likely to import and metabolize sucrose. It is also worth noting that since the original goal of this study was to determine if intravaginal sucrose treatment could improve clinical markers and increase the abundance of *Lactobacillus*, we excluded animals with low vaginal pH and potentially high *Lactobacillus* abundance. In future studies, these animals will be identified and studied longitudinally to determine if this represents a stable or transient community state.

It has been well established that humans in the developing world have a distinct microbiome from those of the developed nations, especially in the context of the gut microbiome (47). For example, individuals in the developing world harbor a more diverse gut microbiome colonized by microbes that have all but disappeared in the Western microbiome (48). This is due to a combination of key environmental factors, such as diet and antibiotic use, among other factors (49–51). Similarly, women in the developing world have a much higher prevalence of diverse non-*Lactobacillus*-dominated communities compared to women in the developed world (4, 29).

As we have previously shown for the gut microbiome, we found that the rhesus VM is also more reflective of women in the developing world (52). Specifically, the data presented here show that macaques harbor a diverse vaginal microbial community similar to that detected in women with a non-*Lactobacillus*-dominated community or symptomatic BV at the 16S rRNA gene amplicon level. Additionally, genomes assembled from the rhesus macaque vaginal microbiome are closely related to human pathobionts associated with BV. However, more work is needed to define the rhesus vaginal microbiome at the strain level to truly understand its value as a preclinical model. Future studies should also focus on the immunological landscape of the rhesus urogenital tract to determine if this diverse community state results in local inflammation, as seen in humans with BV and asymptomatic women with diverse communities. There are many unanswered questions about why this diverse vaginal community is maintained in some women. While this community type has been shown to increase the risk for some sexually transmitted diseases, such as HIV and human papillomavirus

(HPV), it may be important for defense against pathogens that are more prevalent in the developing world. Rhesus macaques offer a unique opportunity to explore the importance of a diverse vaginal community that is highly prevalent in women on a global scale.

## MATERIALS AND METHODS

**Sample collection and cohort information.** All macaque studies were reviewed and approved by the OHSU/ONPRC Institutional Animal Care and Use Committees (IACUC). The animals were socially housed indoors at the Oregon National Primate Research Center (ONPRC) following standards established by the U.S. Federal Animal Welfare Act and the National Research Council's *Guide for the Care and Use of Laboratory Animals*, 8th ed. (National Academies Press, Washington, DC, 2011). All animals were tested annually for simian viruses (simian immunodeficiency virus, simian retrovirus 2, *Macacine alphaherpesvirus 1*, and simian T lymphotropic virus) and received a mammalian old tuberculin test semiannually. The monkeys underwent preassignment evaluations prior to initiation of the study. Qualifications of assignment to the study included normal menstrual cycle, age 5 to 8 years, and a healthy weight of 4 to 8 kg, and all monkeys had to be void of gastrointestinal issues and antibiotic use for greater than 3 months. At the preassignment screening, Nugent score and microbiome samples were collected.

Samples were collected for 16S rRNA amplicon sequencing, metagenomics, and Nugent scoring. Samples used for Nugent scoring, including pH analysis and whiff test, were collected using polyester swabs (Fisher Scientific; item 23-400-122). Samples used for microbiome analysis were collected with Copan swabs (Fisher Scientific; 23-600-957) and stored in 20% glycerol (for shotgun metagenomics) or immediately snap-frozen upon collection and stored at −80°C until DNA extraction (for 16S rRNA amplicon sequencing). To collect the vaginal and rectal swab samples, animals were sedated and placed in a ventral recumbency with their pelvis slightly elevated. A sterile nasal speculum was inserted vaginally to ensure a mid-vaginal sample collection without rectal or vaginal introitus contamination. Preliminary screening identified 3/20 (15%) of adult rhesus macaque females with vaginal pH 4, absence of clue cells, and a negative whiff test. These animals were not included in the longitudinal study. The animals chosen for further study had high pH ($>5$) and other characteristics of VM dysbiosis.

To minimize the natural variability that may occur over the menstrual cycle, animals received 21 days of combination oral contraceptives (Portia; Teva Pharmaceuticals USA, Inc.) to synchronize the group. One day after the last dose of oral contraceptives, samples were collected (Nugent scores, clue cells, whiff test, microbiome, and metagenomics), and animals menstruated approximately 24 to 48 h later. Next, animals underwent postmenstruation sample collection (Nugent scores, clue cells, whiff test microbiome, metagenomics), and then began 5 days of sucrose/placebo gel treatment administration.

Animals were randomly designated to one of two treatment groups: control (vehicle only; $n = 8$) or sucrose (10% sucrose; $n = 8$). The sucrose treatment consisted of 1.6% xanthan gum (Xantural 180; CP Kelco) added to a 10% sucrose solution (Sigma-Aldrich) to create a gel property and brought to a pH of 5.0 using lactic acid (Fisher Scientific). The control treatment did not contain sucrose. Vaginal administration of the gel occurred daily using a 5-ml slip-tip syringe, with control animals receiving the xanthan gum gel only.

**Hormone quantification.** Serum concentrations of estradiol ($E_2$) and progesterone ($P_4$) were assayed by the ONPRC Endocrine Technologies Core. Hormone concentrations were determined using a chemiluminescence-based automated clinical platform (Roche Diagnostics Cobas e411; Roche Diagnostics, Indianapolis, IN). Serum $E_2$ and $P_4$ assay ranges were between 5 and 3,000 pg/ml and 0.05 to 60 ng/ml, respectively. $E_2$ and $P_4$ intra-assay coefficients of variation (CV) were 8.3 and 4.2%, whereas interassay CV were 5.9 and 6.3%, respectively.

**Clinical data generation.** Nugent's scoring was carried out by a trained microbiologist with 4 years of experience performing Nugent scores in a CLIA-approved clinical laboratory using established criteria for clinical studies (53). To complement Nugent scoring, three of four Amsel criteria (54) were also assessed, including vaginal pH, presence of clue cells, and whiff test. Vaginal discharge, the fourth Amsel criteria, was not measured. The presence of clue cells (i.e., vaginal epithelial cells coated with bacteria, resembling a "sandy" appearance) was determined, and samples with ≥20% clue cells were considered positive for this test. Whiff tests were performed by adding 10% KOH to a sample of vaginal fluid, and the presence of a fishy odor was interpreted as a positive test, while its absence was determined as a negative test result. A full breakdown of clinical measurements can be found in Table S1 in the supplemental material.

**16S rRNA gene library construction and sequencing.** Total DNA was extracted from vaginal swabs using the Mo Bio Blood and Tissue DNA Isolation kit (Mo Bio Laboratories, Carlsbad, CA, USA). Rectal swabs were extracted using the PowerSoil DNA Isolation kit (Mo Bio Laboratories, Carlsbad, CA, USA). This DNA was used as the template to amplify the hypervariable V4 region of the 16S rRNA gene using PCR primers (515F/926R, with the forward primer containing a 12-bp barcode) in duplicate reaction mixtures containing 12.5 $\mu$l GoTaq master mix, 9.5 $\mu$l nuclease-free $H_2O$, 1 $\mu$l template DNA, and 1 $\mu$l 10 $\mu$M primer mix. Thermal cycling parameters were 94°C for 3 min, followed by 35 cycles of 94°C for 45 s, 50°C for 1 min, and 72°C for 1 min and 30 s, followed by 72°C for 10 min. PCR products were purified using a MinElute 96 UF PCR purification kit (Qiagen, Valencia, CA, USA). Libraries were sequenced (2 × 300 bases) using Illumina MiSeq.

**16S rRNA gene sequence processing.** Raw FASTQ 16S rRNA gene amplicon sequences were uploaded and processed using the QIIME 2 version 2019.10 (55) analysis pipeline as we have previously

described (52). Briefly, sequences were demultiplexed and quality filtered using the DADA2 plugin for QIIME 2 (56), which filters chimeric sequences. The generated sequence variants were then aligned using MAFFT (57), and a phylogenetic tree was constructed using FastTree 2 (58). Taxonomy was assigned to sequence variants using q2-feature-classifier against the SILVA database (release 119) (59). To prevent sequencing depth bias, samples were rarified to 10,000 sequences per sample before $\alpha$ and $\beta$ diversity analysis. QIIME 2 was also used to generate the following $\alpha$ diversity metrics: richness (as observed taxonomic units), Shannon evenness, and phylogenetic diversity. $\beta$ diversity was estimated in QIIME 2 using weighted and unweighted UniFrac distances (60).

Analysis of the unweighted Unifrac distance (based on the presence/absence of microbes) revealed that 9 vaginal samples were very similar to fecal samples (see Fig. S1A in the supplemental material). These 9 samples had a significantly higher number of observed amplicon sequence variants (ASVs) than the group average of all other vaginal samples (Fig. S1B) and a high relative abundance of taxa typically found in the fecal microbiome (Fig. S1C). Therefore, these nine VM samples were removed from future analysis, along with eight additional samples that did not meet our minimum sequencing depth threshold of 10,000 reads (Fig. S1D).

16S rRNA gene amplicon sequencing data obtained from vaginal swabs from 236 women from Umlazi, South Africa, aged 18 to 23 years, were obtained from Gossman et al. (4). These samples were imported into QIIME 2 and rarified to 13,000 reads per sample. Taxonomy was assigned using the full-length SILVA database (release 119) at the 99% operational taxonomic unit (OTU) cutoff. Genus-level (L6) taxonomy tables were merged, and Bray-Curtis dissimilarity matrices were generated using QIIME 2.

**Shotgun metagenomics.** Shotgun metagenomic libraries were prepared for vaginal samples obtained from all animals at the initiation of oral contraceptives using the Illumina Nextera Flex library prep kit, per the manufacturer's recommended protocol, and sequenced on an Illumina HiSeq 4000 ($2 \times 100$ bases). Raw demultiplexed reads were quality filtered using Trimmomatic (61), and potential host reads were removed by aligning trimmed reads to the *Macaca mulatta* genome (Mmul 8.0.1) using BowTie2 (62). After quality filtering and decontamination, an average of 6.08 million reads (minimum, 0.517 read; maximum, 30 million reads) per sample were used for downstream analysis. Samples with less than 1 million reads after quality filtering were excluded from downstream analysis. Trimmed and decontaminated reads were then annotated using the HUMAnN2 pipeline using the default setting with the UniRef50 database and assigned to GO terms (63). An average of 37.5% (minimum, 20.88%; maximum, 46.42%) of quality-filtered reads were functionally annotated using HUMAnN2. Functional annotations were normalized using copies per million (CPM) reads before statistical analysis (64–66). Taxonomy was assigned to trimmed and decontaminated reads using MetaPhlAn2 within the HUMAnN2 pipeline (67).

Trimmed and decontaminated reads were assembled into contigs using meta-SPAdes with default parameters (68). Assembled contigs of $<1$ kb were also binned into putative metagenomically assembled genomes (MAGs) using MetaBat (69). Genome completeness/contamination was tested using CheckM (70), and all bins with a completeness of $>80\%$ and contamination of $<2\%$ were annotated using PATRIC (71). The taxonomy of draft genomes was determined using PATRIC's similar genome finder.

Shotgun metagenomic sequencing data were obtained from Lev-Sagie et al. (15) and Oliver et al. (28). Human samples were classified as "*Lactobacillus*-dominated" samples if the relative abundance of *Lactobacillus* species was $>90\%$. Samples classified as "diverse human (BV symptomatic)" were the pre-transplant samples from Lev-Sagie et al. (15). Samples classified as "diverse human (asymptomatic)" were obtained from Oliver et al. (28), with a *Lactobacillus* relative abundance of $<90\%$. To avoid pseudoreplication, only samples from the earliest time point were used. Sequences were annotated using the HUMAnN2 pipeline as described above for rhesus vaginal samples. For these human samples, an average of 58.6% (minimum, 34.91%; maximum, 78.02%) of quality-filtered reads were functionally annotated.

**Statistical analysis.** All statistical analyses were conducted used PRISM (V8). QIIME 2 was used to calculate $\alpha$ diversity metrics, observed OTUs, Shannon evenness, and $\beta$ diversity, as well as weighted and unweighted UniFrac distances. Bray-Curtis dissimilarity matrices were constructed for species-level relative abundance. Unpaired $t$ tests or nested $t$ tests when noted and one-way analysis of variance (ANOVA) with *post hoc* correction were implemented using PRISM (V8) to generate $P$ values. Correlations between microbes and clinical measurements were generated in R using the rmcorr package, which calculates correlations within individuals (72). The LEfSe algorithm was used to identify differentially abundant taxa and pathways between groups with a logarithmic linear discriminant analysis (LDA) score cutoff of 2 (66).

**Data availability.** The data sets generated for this study are available in the NCBI SRA repository, under bioproject ID no. PRJNA704084. This bioproject includes 16S amplicon sequences, unassembled shotgun metagenomic sequences, and metagenomically assembled genomes.

## SUPPLEMENTAL MATERIAL

Supplemental material is available online only.

**FIG S1**, EPS file, 2 MB.
**FIG S2**, EPS file, 2 MB.
**FIG S3**, EPS file, 1.8 MB.
**FIG S4**, EPS file, 1.5 MB.

**FIG S5**, PDF file, 0.6 MB.
**TABLE S1**, PDF file, 0.1 MB.
**TABLE S2**, PDF file, 0.1 MB.

## ACKNOWLEDGMENTS

This work was supported by the Bill and Melinda Gates Foundation (OPP1159806).
N.S.R. was supported by NIH/NIAID T32 training grants (AI141346 and AI007319).

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
