## [Reviewer comments · mSystems]

Longitudinal profiling of the macaque vaginal microbiome reveals similarities to diverse human vaginal communities.

Nicholas Rhoades, Sara Hendrickson, Danielle Gerken, Kassandra Martinez, Ov Slayden, Mark Slifka, and Ilhem Messaoudi

Corresponding Author(s): Ilhem Messaoudi, University of California, Irvine

Review Timeline:

Submission Date:	December 16, 2020
Editorial Decision:	January 30, 2021
Revision Received:	March 1, 2021
Accepted:	March 24, 2021

Editor: Thomas Sharpton

Reviewer(s): The reviewers have opted to remain anonymous.

Transaction Report:

DOI: <https://doi.org/10.1128/mSystems.01322-20>

January 30, 2021

Prof. Ilhem Messaoudi
University of California, Irvine
UC Irvine Molecular Biology and Biochemistry
Irvine, CA

Re: mSystems01322-20 (Longitudinal profiling of the macaque vaginal microbiome reveals similarities to diverse human vaginal communities: implications for use as a pre-clinical model for bacterial vaginosis.)

Dear Prof. Ilhem Messaoudi:

Thank you for submitting your manuscript to mSystems. Three reviewers have assessed the manuscript and suggest several revisions prior to its publication. I invite you to consider these suggestions and submit a revised manuscript. In addition to accounting for reviewer remarks, please ensure that your revised manuscript complies with the mSystems Data Availability requirements.

Below you will find the comments of the reviewers. Please let me know if you have any questions.

To submit your modified manuscript, log onto the eJP submission site at <https://msystems.msubmit.net/cgi-bin/main.plex>. If you cannot remember your password, click the "Can't remember your password?" link and follow the instructions on the screen. Go to Author Tasks and click the appropriate manuscript title to begin the resubmission process. The information that you entered when you first submitted the paper will be displayed. Please update the information as necessary. Provide (1) point-by-point responses to the issues raised by the reviewers as file type "Response to Reviewers," not in your cover letter, and (2) a PDF file that indicates the changes from the original submission (by highlighting or underlining the changes) as file type "Marked Up Manuscript - For Review Only."

Due to the SARS-CoV-2 pandemic, our typical 60 day deadline for revisions will not be applied. I hope that you will be able to submit a revised manuscript soon, but want to reassure you that the journal will be flexible in terms of timing, particularly if experimental revisions are needed. When you are ready to resubmit, please know that our staff and Editors are working remotely and handling submissions without delay. If you do not wish to modify the manuscript and prefer to submit it to another journal, please notify me of your decision immediately so that the manuscript may be formally withdrawn from consideration by mSystems.

Sincerely,

Thomas Sharpton

Editor, mSystems

Journals Department
Reviewer comments:

Reviewer #1 (Comments for the Author):

See attached

Reviewer #2 (Comments for the Author):

A relevant model of the human vaginal microbiome is a valuable and much needed tool for research in this area. Rhoades et al. present good evidence, at least in part, for the use of rhesus macaques as a tractable model. The importance and rationale for the paper is well covered, with a good, broad perspective taken, clear methods, and sound discussion and conclusions based on the results presented.

There are some minor comments that need addressing or clarification.

Line 329: States that animals were sampled at 5 timepoints, however later on Line 350, it says animals were measured across 8 timepoints. This is confusing. Please clarify.

Line 360: It really needs to be clearer throughout the manuscript how many samples were analysed. The 120 value here does not seem to make sense based on 16 animals sampled across 8 timepoints. It is mentioned in the methods that some samples did not meet a quality inclusion step, but of the 17 of these, how many were samples for the initial analysis and how many were from the trial. Furthermore, this value of 120 does not seem to agree with line 409-410. I suggest that rather than emphasise the samples that were removed, please be clearer as to the number of samples analysed.

Line 357: Where is this data presented?

Line 360-363: You state that 13/120 samples were dominated by a single microbe, but then you list 15 samples in the taxa breakdown, please clarify.

Line 378: "Progesterone" does not need to be capitalised

Line 391: There is repetition of "differences" in this sentence

Line 475: This statement that "...50% of women globally have a diverse vaginal microbiome..." is a broad generalisation and if truly speaking globally, it is based on studies from limited geographical spread (primarily South Africa and USA). Please rephrase to reflect the limited global data, or provide stronger evidence to the claim.

Line 511-513: It is important to note in the discussion, that this study selectively omitted animals that did not present with classical BV type symptoms (line 180). In doing so there may be a bias against animals with a more lactobacilli dominant VM? Please consider in your discussion.

Figure 2: The legend does not correctly match the panels in the figure, particularly panel E. What do the individual columns in this panel E heatmap represent? What does the different colours for the tree branches mean in panel A?

Figure 5: The font size on the labels of panel B need to be increased.

Supplemental Figure 4: The figure legend does not match the content of the figure. There appears to be an additional panel that is not described. It is also not possible to read the GO terms on the heatmaps. Please improve the resolution

Reviewer #3 (Comments for the Author):

The manuscript by Rhoades, Hendrickson, et al reports a longitudinal study of the vaginal microbiome in Rhesus macaque, and the potential microbiome response of a subgroup to a sucrose treatment.

The research was reviewed and approved by the Institutional Animal Care and Use Committee (IACUC) and appears to follow Federal standards for animal welfare.

The methods include sufficient details to enable reproducibility and are appropriate for answering the authors research questions. Results and conclusions such as taxonomic stability over time and individuals being the strongest factors affecting microbiome diversity support prior observations in human vaginal microbiomes.

Although the number of individuals investigated is small, the manuscript addresses interesting comparisons between the vaginal microbiome of rhesus macaque and humans, it is well written, the story flows well, and provides support for the authors suggestion that Rhesus macaques are good model systems for therapeutic intervention research that could be translated to humans.

One concern is the missing "Data Availability" section in the manuscript. Authors should deposit their sequences (16S reads, metagenomic reads, and assembled genomes) in a public repository such as GenBank and link them back to the manuscript.

Minor comments:

1. Page 11 line 236. I suggest mentioning that the DADA2 plugin in the Qiime2 pipeline was used for pre-processing reads (the way it reads now appears to us DADA2 independently and then using the Qiime2 pipeline for analysis).

2. Page 12 line 270 & page 13 line 286. Please include information about how many metagenomic reads were included in the final HUMAnN2 predictions. These numbers are useful to readers to estimate the amount of data used in the analysis vs. amount of data originally generated.
3. Please include error bars on Figs. S2 and S3 (it would be easier to see the spread)
4. Page 20 line 452-471. It would be helpful to know a more general context for the pathways mentioned. For example, pathways involved in carbohydrate metabolism are enriched in X vs. Y.
5. Page 22 line 506. It has been shown that there are likely between 4 and 5 *Gardnerella* sub-types in human vaginal microbiomes. Do the authors know if their assembled *Gardnerella* genomes belong to any of those sub-groups?
6. Page 23 line 528. In fact, stability in the vaginal microbiome has been observed in pregnant and non-pregnant women, although transitions between states do occur (see for example Romero et al 2014, DiGiulio et al 2015, and others).
7. Figure 3 legend. The authors correctly reported in the Results section that due to potential confounding effects of different laboratory methodologies, any comparisons between the macaque VM and human VM from other studies would be made qualitatively. Therefore, I would suggest to remove the significant p-values reported in the legend of Figure 3, because significant p-values cannot accurately be attributed solely to microbiome differences and similarities between communities. Unless differences in methodology were added as potential factors in the statistical models.
8. Figure 5 legend. Can the authors clarify if the significant p-values in Figure 5C correspond to direct human-human VM comparisons? If they are meant to represent significant differences/similarities between human and macaque VM, then I would suggest not to include p-values, unless the experiments were all done by the same authors, or any methodological or other potential confounding effects were considered in the statistical models.
9. Supplemental Figure 1 legend. Again, I would suggest to remove significant p-values from the legend.

Comments: This manuscript examines the vaginal microbiome of Rhesus macaques and compares the results to the human vaginal microbiome. It presents important findings relevant to the primate's microbiome indicating and confirming that it is more diverse than the normal human vaginal microbiome. The manuscript attempts to correlate the macaque microbiome to the microbiome exhibited by women with bacterial vaginosis. This relationship is less than clear. Although there is overlap of genera between the two target sites, there is little demonstrated overlap of species or strains. This is somewhat addressed by a functional analysis of metagenomic sequences; i.e., the BV microbiome tends to cluster more closely to the primate microbiome. Yet, the idea that the macaque can be used as a proxy for the human microbiome is more than a bit tenuous. Overall, the manuscript presents excellent data but probably should tone down the claims of using the macaque as a proxy for humans in this area.

1. The samples showed near absence of bacteria considered hallmarks of BV: *G. vaginalis* and BVAB1, and other BV-associated taxa. Hence, considering these macaques as a 'good' model for BV is controversial. Interestingly, *Mobiluncus* taxa are common and prevalent in these samples. Most recent studies have indicated that *Mobiluncus* is rarely a dominant taxon in women with BV. Moreover, characterizing a vaginal microbiome at the genus level, as indicated in Table 1, neglects the diversity of these genera. There are plentiful examples in the literature showing species diversity, not to mention strain diversity. It's not clear why Table 1 is limited to only genus level identifications. Also, *L. iners* and *L. crispatus* have different impacts on the concept of BV, and bundling them under the title of 'Lactobacillus' is possibly misleading. I suggest truncating the title to "Longitudinal profiling of the macaque vaginal microbiome reveals similarities to diverse human vaginal communities".
2. It seems a bit strange to describe a model for the human vaginal microbiome as occasionally dominated by 'less common microbes such as *Gardnerella* and *Lactobacillus*', which are generally two of the most common components of the human vaginal microbiome. Similarly, it is a bit unusual for human vaginal microbiomes NOT to be dominated by a single taxon. As a model of BV, the general paucity of *Gardnerella* is striking.

354 pH (Figure 2 A, C). The VM of the remaining 9 animals was more variable, transitioning
355 between states and communities dominated by less common microbes such as
356 *Gardnerella*, and *Lactobacillus* (Figure 2C). Finally, we found that individual

3. The data in Figure 3 are not convincing. Panel A shows a PCA of the Bray Curtis distances between samples. Whereas the different types of samples seem to cluster, it is not clear that the diverse human and macaque samples cluster closely. The *Lactobacillus* dominated samples may be more convincing, but there were few of these in the macaque group. Fig. 3C is not clear. What are the colors?
4. On line 413, the manuscript identifies *L. johnsonii*, *L. amylovorus*, and *L. acidophilus* in four of the macaque samples. These are less common in the human vaginal microbiome than others. Were these the only species of Lacto identified?
5. Figure 4 is most interesting in that the macaque isolates generally seem to cluster together, apart from the known human isolates. For the most part anyway. This suggests an evolution away from the human system (or, more relevantly, evolution of the human strains away from the primate strains). It should be noted that *G. vaginalis* has been speciated into four major groups and up to 13 species. It's not clear from the text or figure which of these is being used in the analysis in Fig. 4. It would be interesting if one or more were more closely related to the macaque strains.

6. It needs to be acknowledged that even though there is some similarity between the BV microbiome and the microbiomes of macaques, this really needs to be examined at the species/strain level. The functional data provided is intriguing, but the overlap between the two VMs is more than a bit tenuous and not entirely convincing. For example, *G. vag* is present in almost all BV samples, but is only relatively poorly represented in macaques, and the actual species/strains are not the same (or at least not demonstrably so).
7. A final concern is that there were a modest number of animals providing longitudinal samples. In most cases, data from these samples was presented as independent. However, it would be a mistake to call longitudinal samples from the same animal independent, even more so since the animals were hormonally synchronized prior to the study. The authors should consider this more carefully in their analyses.

Minor comments

1. The correlation of BV and inflammation is controversial:

100 This diverse community is often associated with a heightened inflammatory state,
 101 increased susceptibility to sexually transmitted diseases notably HIV, and the
 102 development of bacterial vaginosis (BV) (7).
 103 BV is an inflammatory disorder and the most common vaginal infection in the US
 103 BV is an inflammatory disorder and the most common vaginal infection in the US

2. Fix sentence: line 92

Lactobacilli inhibit the growth of other vaginal microbes via the production of lactic acid
 through fermentation which decreases the pH of the vaginal environment (pH 3.5-5.5)
 inhibiting the growth of other microbes (3). *Lactobacillus* species also competitively

3. Sneathia is spelled as Snethia throughout? I thought it should be Sneathia...
4. I could not find figure 2F...maybe 2E?
5. On Fig 2A, why are there more than onee Treponeme?
6. What is meant by 'genome assembly' in the following:

411 Although we were unable to identify the *Lactobacillus* colonizing the rhesus VM at the
 412 species level using 16S amplicon sequencing or genome assembly, annotation of

7. Line 459 refers Figure 5C. Is it rather 5B?
8. 16 S rRNA is often written as 16s throughout the manuscript.
9. Line 522 suggests that *L. acidophilus* hasn't been identified in human samples. That's really an overstatement. It is likely overlooked in many cases because of the similarity of its rRNA sequence to that of other *Lactobacilli* (e.g. *L. crispatus*).

We thank the reviewers for their thoughtful and thorough reviews of our manuscript. We have edited the text and figures in response to the concerns they raised (indicated by yellow highlighting) and provide detailed responses below.

Reviewer #1 (Comments for the Author):

1. The samples showed near absence of bacteria considered hallmarks of BV: *G. vaginalis* and BVAB1, and other BV-associated taxa. Hence, considering these macaques as a 'good' model for BV is controversial.

BVAB was not detected within vaginal microbial communities in these monkeys. We have toned down our statements about rhesus macaques being used as an animal model for BV throughout the manuscript and focused our manuscript on the diverse community state type.

Interestingly, *Mobiluncus* taxa are common and prevalent in these samples. Most recent studies have indicated that *Mobiluncus* is rarely a dominant taxon in women with BV.

While *Mobiluncus* taxa were prevalent within our study population, they were not dominant in any samples. It was only detected with a 25 and 20% relative abundance in two samples, but for most samples it was detected with a relative abundance <5%. This seems to be in agreement with the human data.

Moreover, characterizing a vaginal microbiome at the genus level, as indicated in Table 1, neglects the diversity of these genera. There are plentiful examples in the literature showing species diversity, not to mention strain diversity. It's not clear why Table 1 is limited to only genus level identifications.

We are unable to improve the taxonomic resolution (Table 1) since 16S data generated from rhesus macaques can often only be classified to the genus or higher level. This is likely due to: 1) as shown in figure 4, the bacteria of the rhesus microbiome often are distinct from their human counterparts at the genomic level and therefore do not fall into the same species at the 16S level; 2) the databases used to assign taxonomy for 16S amplicon data lack sequences from macaque associated bacteria and are heavily biased towards known human associated species. Our exclusion of low pH animals from the longitudinal study and inability to assemble a *Lactobacillus* MAG likely hampered our ability to identify these *Lactobacilli* at the species level and will be targeted in future studies.

Also, *L. iners* and *L. crispatus* have different impacts on the concept of BV, and bundling them under the title of '*Lactobacillus*' is possibly misleading.

We agree and now acknowledge this more clearly in the discussion.

I suggest truncating the title to "Longitudinal profiling of the macaque vaginal microbiome reveals similarities to diverse human vaginal communities".

The title has now been changed

2. It seems a bit strange to describe a model for the human vaginal microbiome as occasionally dominated by ‘less common microbes such as Gardnerella and Lactobacillus’, which are generally two of the most common components of the human vaginal microbiome. Similarly, it is a bit unusual for human vaginal microbiomes NOT to be dominated by a single taxon. As a model of BV, the general paucity of Gardnerella is striking.

354 pH (Figure 2 A, C). The VM of the remaining 9 animals was more variable, transitioning
355 between states and communities dominated by less common microbes such as
356 Gardnerella, and Lactobacillus (Figure 2C). Finally, we found that individual

We have now clarified this sentence to state that these microbes are less common within our study population. We have re-focused our manuscript away from BV and towards the diverse community state type. While Gardnerella was not a dominant taxa, it is relatively abundant. Indeed, it was detected in 62 of 112 samples analyzed by 16S amplicon sequencing, and we were able to assemble a Gardnerella genome from 9 of 21 samples subjected to shotgun metagenomics.

3. The data in Figure 3 are not convincing. Panel A shows a PCA of the Bray Curtis distances between samples. Whereas the different types of samples seem to cluster, it is not clear that the diverse human and macaque samples cluster closely. The Lactobacillus dominated samples may be more convincing, but there were few of these in the macaque group. Fig. 3C is not clear. What are the colors?

This figure represents a limited qualitative analysis because we are unable to account for difference such as sample handling and methodology that may be introducing variability. However, we do believe that including this figure is important to give the reader context for how the rhesus vaginal microbiome compares to humans.

We have added a key to Figure 3C.

4. On line 413, the manuscript identifies *L. johnsonii*, *L. amylovorus*, and *L. acidophilus* in four of the macaque samples. These are less common in the human vaginal microbiome than others. Were these the only species of Lacto identified?

These were the only species identified from our shotgun metagenomic data using short read annotation which is not as precise as genome assembly. We were unfortunately unable to assemble MAG for these Lactobacillus species.

5. Figure 4 is most interesting in that the macaque isolates generally seem to cluster together, apart from the known human isolates. For the most part anyway. This suggests an evolution away from the human system (or, more relevantly, evolution of the human

strains away from the primate strains). It should be noted that *G. vaginalis* has been speciated into four major groups and up to 13 species. It's not clear from the text or figure which of these is being used in the analysis in Fig. 4. It would be interesting if one or more were more closely related to the macaque strains.

We have now added all 13 “genomic species” identified in humans to Fig. 4, all of which are still distinct our assembled Gardnerella.

6. It needs to be acknowledged that even though there is some similarity between the BV microbiome and the microbiomes of macaques, this really needs to be examined at the species/strain level. The functional data provided is intriguing, but the overlap between the two VMs is more than a bit tenuous and not entirely convincing. For example, *G. vag* is present in almost all BV samples, but is only relatively poorly represented in macaques, and the actual species/strains are not the same (or at least not demonstrably so).

We acknowledge this limitation more fully in the discussion.

7. A final concern is that there were a modest number of animals providing longitudinal samples. In most cases, data from these samples was presented as independent. However, it would be a mistake to call longitudinal samples from the same animal independent, even more so since the animals were hormonally synchronized prior to the study. The authors should consider this more carefully in their analyses.

We have updated our statistical analysis to account for the repeated measures of our study design with the exception of Figure 3 which is largely a qualitative analysis. Specifically, Figure 1 G and H have been changed to nested T-tests and Figure 2 E Now uses repeated measures correlations.

Minor comments

1. The correlation of BV and inflammation is controversial:

We have removed inflammation from this statement.

2. Fix sentence: line 92

Lactobacilli inhibit the growth of other vaginal microbes via the production of lactic acid through fermentation which decreases the pH of the vaginal environment (pH 3.5-5.5) inhibiting the growth of other microbes (3). *Lactobacillus* species also competitively

This has been fixed

3. *Sneathia* is spelled as *Snethia* throughout? I thought it should be *Sneathia*...

We have fixed this mistake throughout the text

4. I could not find figure 2F...maybe 2E?

Apologies for this confusion, it should be Figure 2E.

5. On Fig 2A, why are there more than one Treponeme?

Sorry this was an artifact of taxonomic classification. Some Treponema was classified as the genus “Treponema” while other sequences were classified as “Treponema (uncultured bacteria)”. The uncultured bacteria was truncated at some point but has now been added back to the figure.

6. What is meant by ‘genome assembly’ in the following:

411 Although we were unable to identify the *Lactobacillus* colonizing the rhesus VM at the

412 species level using 16S amplicon sequencing or genome assembly, annotation of

Updated to state that this is referring to metagenomic genome assembly.

7. Line 459 refers Figure 5C. Is it rather 5B?

It is indeed 5B, Fixed.

8. 16 S rRNA is often written as 16s throughout the manuscript.

This has been corrected.

9. Line 522 suggests that *L. acidophilus* hasn’t been identified in human samples. That’s really an overstatement. It is likely overlooked in many cases because of the similarity It is likely overlooked in many cases because of the similarity of its rRNA sequence to that of other Lactobacilli (e.g. *L. crispatus*).

We have now clarified this statement in text. We have acknowledged the relatedness of *L. acidophilus* to *L. crispatus*, and eluded to the fact that more data at the genome level is needed to resolve this question.

Reviewer #2 (Comments for the Author):

1. Line 329: States that animals were sampled at 5 timepoints, however later on Line 350, it says animals were measured across 8 timepoints. This is confusing. Please clarify.

We have clarified that all animals were sampled at 8 timepoints. The 5 timepoint statement was in reference to post-treatment samples.

2. Line 360: It really needs to be clearer throughout the manuscript how many samples were analysed. The 120 value here does not seem to make sense based on 16 animals sampled across 8 timepoints. It is mentioned in the methods that some samples did not meet a quality inclusion step, but of the 17 of these, how many were samples for the initial analysis and how many were from the trial. Furthermore, this value of 120 does not seem to agree with line 409-410. I suggest that rather than emphasise the samples that were removed, please be clearer as to the number of samples analysed.

We have added additional information to Figure 1A to clarify how many samples were used at each timepoint, and updated all numbers in text.

3. Line 357: Where is this data presented?

We have added a new Supp. Figure 4, where we present the raw data associated with these statistics.

4. Line 360-363: You state that 13/120 samples were dominated by a single microbe, but then you list 15 samples in the taxa breakdown, please clarify.

The actual number was 15/112. This has been fixed throughout the manuscript.

5. Line 378: "Progesterone" does not need to be capitalized

Fixed

6. Line 391: There is repetition of "differences" in this sentence

Fixed

7. Line 475: This statement that "...50% of women globally have a diverse vaginal microbiome..." is a broad generalisation and if truly speaking globally, it is based on studies from limited geographical spread (primarily South Africa and USA). Please rephrase to reflect the limited global data, or provide stronger evidence to the claim.

We rephrased this statement and added studies from Asia, south America, and Europe.

8. Line 511-513: It is important to note in the discussion, that this study selectively omitted animals that did not present with classical BV type symptoms (line 180). In doing so there may be a bias against animals with a more lactobacilli dominant VM? Please consider in your discussion.

We have added this caveat to our discussion.

9. Figure 2: The legend does not correctly match the panels in the figure, particularly panel E. What do the individual columns in this panel E heatmap represent? What does the different colours for the tree branches mean in panel A?

The heatmap represent correlation values between the 25 microbes in the large heatmap and indicated clinical measurements. This has now been added to the figure legend,

10. Figure 5: The font size on the labels of panel B need to be increased.

The font size for the labels has been increased.

11. Supplemental Figure 4: The figure legend does not match the content of the figure.

There appears to be an additional panel that is not described. It is also not possible to read the GO terms on the heatmaps. Please improve the resolution

This is now Supp. Figure 5: We have re-written the figure legend, and increased the size of the GO term labels.

Reviewer #3 (Comments for the Author):

1. One concern is the missing "Data Availability" section in the manuscript. Authors should deposit their sequences (16S reads, metagenomic reads, and assembled genomes) in a public repository such as GenBank and link them back to the manuscript.

A data availability section has now been added to the manuscript. All raw sequencing data has been deposited in the NCBI SRA under the BioProject: PRJNA704084. Verification of this submission can be found at the following link:
<https://dataview.ncbi.nlm.nih.gov/object/PRJNA704084?reviewer=vojmuf74tadsrltrcel7of72mc>

Minor comments:

1. Page 11 line 236. I suggest mentioning that the DADA2 plugin in the Qiime2 pipeline was used for pre-processing reads (the way it reads now appears to us DADA2 independently and then using the Qiime2 pipeline for analysis).

This has now been clarified in the text.

2. Page 12 line 270 & page 13 line 286. Please include information about how many metagenomic reads were included in the final HUMAnN2 predictions. These numbers are useful to readers to estimate the amount of data used in the analysis vs. amount of data originally generated.

We have added the % of reads functionally annotated for both rhesus macaque and human samples.

3. Please include error bars on Figs. S2 and S3 (it would be easier to see the spread)

Figures S2 and S3 have been changed to bar graphs with error bars and individual point to better represent the data. Figure S2- panels C and D are showing the % of animals that are positive for a particular clinical measure and therefore cannot have error bars. We have relabeled the Y axis of these graphs to make this more clear.

4. Page 20 line 452-471. It would be helpful to know a more general context for the pathways mentioned. For example, pathways involved in carbohydrate metabolism are

enriched in X vs. Y.

We have added additional information about the pathways mentioned in text.

5. Page 22 line 506. It has been shown that there are likely between 4 and 5 Gardnerella sub-types in human vaginal microbiomes. Do the authors know if their assembled Gardnerella genomes belong to any of those sub-groups?

Thank you for this comment, an additional review also mentioned that Gardnerella has been split into 13 “Genomic species”. We have now included all 13 of these species (which encompass the sub-types you mentioned) and show that our assembled genomes are in fact distinct. (See Figure 4C)

6. Page 23 line 528. In fact, stability in the vaginal microbiome has been observed in pregnant and non-pregnant women, although transitions between states do occur (see for example Romero et al 2014, DiGiulio et al 2015, and others).

We have added this information and expanded on this topic within the discussion.

7. Figure 3 legend. The authors correctly reported in the Results section that due to potential confounding effects of different laboratory methodologies, any comparisons between the macaque VM and human VM from other studies would be made qualitatively. Therefore, I would suggest to remove the significant p-values reported in the legend of Figure 3, because significant p-values cannot accurately be attributed solely to microbiome differences and similarities between communities. Unless differences in methodology were added as potential factors in the statistical models.

The p-values have been removed from this figure.

8. Figure 5 legend. Can the authors clarify if the significant p-values in Figure 5C correspond to direct human-human VM comparisons? If they are meant to represent significant differences/similarities between human and macaque VM, then I would suggest not to include p-values, unless the experiments were all done by the same authors, or any methodological or other potential confounding effects were considered in the statistical models.

The p-values have been removed from this figure.

9. Supplemental Figure 1 legend. Again, I would suggest to remove significant p-values from the legend.

These data, including fecal samples, were collected process and analyzed by us. As a part of the larger vaginal dataset and therefore can be statistically compared.

March 24, 2021

Prof. Ilhem Messaoudi
University of California, Irvine
UC Irvine Molecular Biology and Biochemistry
Irvine, CA

Re: mSystems01322-20R1 (Longitudinal profiling of the macaque vaginal microbiome reveals similarities to diverse human vaginal communities.)

Dear Prof. Ilhem Messaoudi:

Thank you for submitting your revised manuscript to mSystems. Based on the feedback of the manuscript referees as well as my own assessment of the revisions, I am happy to recommend your manuscript for publication in mSystems. Please see the following information about next steps and do let me know if you have any questions. Congratulations!

Your manuscript has been accepted, and I am forwarding it to the ASM Journals Department for publication. For your reference, ASM Journals' address is given below. Before it can be scheduled for publication, your manuscript will be checked by the mSystems senior production editor, Ellie Ghatineh, to make sure that all elements meet the technical requirements for publication. She will contact you if anything needs to be revised before copyediting and production can begin. Otherwise, you will be notified when your proofs are ready to be viewed.

- Minimum resolution of 1280 x 720
- .mov or .mp4. video format
- Provide video in the highest quality possible, but do not exceed 1080p
- Provide a still/profile picture that is 640 (w) x 720 (h) max

We recognize that the video files can become quite large, and so to avoid quality loss ASM suggests sending the video file via <https://www.wetransfer.com/>. When you have a final version of the video and the still ready to share, please send it to Ellie Ghatineh at eghatineh@asmusa.org.

Sincerely,

Thomas Sharpton
Editor, mSystems

Journals Department
Fig. S3: Accept
Fig. S4: Accept
Fig. S2: Accept
Fig. S1: Accept
Table S2: Accept
Fig. S5: Accept
Table S1: Accept